# A Pipeline for Constructing Reference Genomes for Large Cohort-Specific Metagenome Compression

**DOI:** 10.3390/microorganisms11102560

**Published:** 2023-10-14

**Authors:** Linqi Wang, Renpeng Ding, Shixu He, Qinyu Wang, Yan Zhou

**Affiliations:** 1State Key Laboratory of Genetic Engineering, School of Life Sciences, Fudan University, Shanghai 200438, China; linqiwang21@m.fudan.edu.cn (L.W.); qinyuwang22@m.fudan.edu.cn (Q.W.); 2MGI Tech, Shenzhen 518083, China; dingrenpeng@genomics.cn (R.D.); heshixu@genomics.cn (S.H.)

**Keywords:** metagenomics, sequence data, reference-based compression

## Abstract

Metagenomic data compression is very important as metagenomic projects are facing the challenges of larger data volumes per sample and more samples nowadays. Reference-based compression is a promising method to obtain a high compression ratio. However, existing microbial reference genome databases are not suitable to be directly used as references for compression due to their large size and redundancy, and different metagenomic cohorts often have various microbial compositions. We present a novel pipeline that generated simplified and tailored reference genomes for large metagenomic cohorts, enabling the reference-based compression of metagenomic data. We constructed customized reference genomes, ranging from 2.4 to 3.9 GB, for 29 real metagenomic datasets and evaluated their compression performance. Reference-based compression achieved an impressive compression ratio of over 20 for human whole-genome data and up to 33.8 for all samples, demonstrating a remarkable 4.5 times improvement than the standard Gzip compression. Our method provides new insights into reference-based metagenomic data compression and has a broad application potential for faster and cheaper data transfer, storage, and analysis.

## 1. Introduction

Metagenomics is one of the most important methods of microbiome research, which uses genomic strategies to investigate the genetic composition and functional patterns of all microorganisms present in specific environmental samples [1]. Over the years, several large-scale collaborative microbiome projects have been initiated, including the well-known Human Microbiome Project (HMP) and the Earth Microbiome Project (EMP) [2,3], the Metagenomics of the Human Intestinal Tract (MetaHIT) for gut microbiota [4], and the TARA Oceans Project for marine microorganisms [5]. These projects usually use non-targeted shotgun sequencing strategies to obtain higher taxonomic resolution and more detailed functional information about the genome, resulting in the generation of massive sequencing data. As of October 2022, the Human Microbiome Project Data Portal (https://portal.hmpdacc.org/) (accessed on 1 October 2022) contains 31,596 samples from 18 studies, with a 48.54 TB file volume of data. The expansion of project size and the growth of data volume poses a challenge in terms of computation, as there may not be enough resources to store and process data. This underscores the importance and urgency of developing methods to compress metagenomic data efficiently.

Several data compression strategies have been developed and can be divided into reference-free and reference-based methods. Reference-free methods compress raw sequencing data based on their natural statistics. Redundant DNA sequences are identified as far as possible and then processed using general compression methods such as Gzip and Bzip2 [6,7,8,9]. Reference-based methods exploit similarities between reads and a reference genome, which usually maps the target sequence to the reference genome and stores the information needed to reconstruct the sequence: position in the reference genome and differences [10,11,12,13,14]. Due to the high similarity among genomes of homologous species, reference-based compression usually achieves high compression ratios, but this approach strongly relies on high-quality reference genomes.

There are many challenges in the reference-based compression of metagenomic sequencing data. Unlike classical genomic samples, metagenomic samples consist of many different organisms, resulting in a lack of universal reference genomes among samples. The microbial composition of cohorts may vary considerably as different studies have various technical approaches to sample collection and sequencing. In addition, microbial reference genomes are often stored in integrated microbial databases, such as the NCBI reference sequence database (RefSeq) [15], iMicrobe [16], EBI Metagenomics [17], IMG/M [18], and MG-RAST [19]. Nevertheless, these databases have extremely large data volumes and a high redundancy of genomes, which inevitably consume significant computational resources and time costs when used as reference genomes for compression.

In this work, we propose a scheme for the construction of lightweight and cohort-specific reference genomes. By overcoming the limitations of existing approaches and utilizing cohort-specific reference genomes, we achieve significant improvements in compression ratios and storage efficiency. Our results on diverse habitats demonstrate the effectiveness and applicability of our pipeline for enhancing the performance of reference-based compression tools and reducing storage costs.

## 2. Materials and Methods

### 2.1. Cohort Description

Studies from January 2015 to October 2022 with paired-end shotgun metagenomic sequencing on human gut, mouth, skin, vaginal, soil, marine, freshwater, and wastewater samples were searched on the National Center for Biotechnology Information (NCBI). In total, 29 studies with a sample size greater than 100 and data sizes greater than 100 GB of published metagenomic datasets were included (Table 1). Two hundred samples were randomly selected from each dataset (all were included if sample size < 200). The details of the total 5669 samples of all datasets are reported in Appendix A. Metagenomic datasets were locally downloaded from the European Nucleotide Archive (ENA) via the Aspera ascp command line client (v3.9.1). Sequencing depth distributions within these datasets are shown in Appendix A.

### 2.2. Data Pre-Processing and Taxonomic Profiling

Metagenomic sequencing reads were trimmed using fastp (v0.23.1) with a minimum N base number of 0 (--n_base_limit 0) and minimum read length of 60 (--length_required 60) for trimming. Taxonomic profiles were obtained using the default parameters of MetaPhlAn3 [49], which uses a database of clade-specific markers to quantify bacteria constituents at the species and higher taxonomic levels.

### 2.3. Data Analysis

To evaluate the heterogeneity of different datasets, we performed a CLR transformation of the species relative abundance data, followed by principal component analysis (PCA) using the scikit-learn module in Python. For each sample, the proportion of species that contributed more than 80% to abundance (abundant species) was first calculated. Next, the richness of the abundant species of various datasets was evaluated using sampling-unit-based rarefaction and extrapolation curves, which were generated using the first (species richness, q = 0) Hill number. The rarefaction/extrapolations were computed using the R package iNEXT (v2.0.12) [50] as the mean of 100 replicate bootstrapping runs to estimate 95% confidence intervals.

### 2.4. Construction of the Basic Reference Database

We chose MetaPhlAn3 as the basis for the construction of our basic reference database, as its marker gene database is the largest and most comprehensive available. Compared with other tools based on marker genes or 16S rRNA, MetaPhlAn3 can detect more microbial species, including some rare or newly discovered ones [49]. We constructed the MetaPhlAan3_db_25k database by selecting the best genomes for each species from NCBI based on the species list of MetaPhlAn3. For each species, we chose representative genomes or reference genomes if available, or otherwise the highest quality assembled genomes. We also downloaded some external genomes, which belong to different clades but also contain MetaPhlAn3 marker genes. These genomes may have acquired marker genes from different species due to horizontal gene transfer or other mechanisms. We believe that these genomes are valuable because they can increase the species diversity and coverage of the basic reference database. For genomes in the form of chromosomes, scaffolds, or contigs, we removed the gaps in the genomes and concatenated all segments into one complete sequence for subsequent alignment and compression. For the MetaPhlAan3_db_25k database, we collected a total of 25,435 microbial genomes, covering various taxa such as bacteria, archaea, and eukaryotes. Similarly, we also constructed the Env_db_6k database, which is a database containing 6149 reference genomes of environmentally relevant microorganisms. Compared with the MetaPhlAan3_db_25k database, the Env_db_6k database contains more environmental microbial species, but it also has some overlaps and crossovers. The two databases can complement each other in reflecting the diversity and variability of the microbial community.

### 2.5. Evaluation of Short-Read Sequence Aligners

Ref1000 and a subset of 10 samples from PRJEB11419 were used to evaluate the performance of aligners during indexing and alignment. Ref1000 is a collection of 1000 bacterial genomes randomly selected from the MetaPhlAn3_db_25k database. Details of Ref1000 and the sample dataset can be found in Appendix A, respectively. We evaluate the time and memory consumption of short-read aligners during indexing using one thread. The BWA-index was run with default parameters because it does not provide a multi-threaded mode.

### 2.6. Construction of Cohort-Specific Reference Genomes and Compression

We randomly selected 50 samples from each dataset and built the cohort-specific reference genomes consisting of the top 1000 genome sequences with the highest alignment rates based on the Met-aPhlAn3_db_25k and Env_db_6k databases. The rest of the samples in each dataset were used for compression tests. The sequencing data of each sample were compressed using Genozip (v13.0.20) under both reference-free and reference-based modes, followed by the calculation of the compression ratio (size of the original FASTQ file/size of the compressed file). The following parameters were used to obtain the best compression: “--best=No_REF” for reference-free mode; “—best” and “—pair” for reference-based mode.

## 3. Results

### 3.1. Microbiota Composition Analysis of Samples from Various Cohorts

We collected 5669 metagenomic samples from 29 public datasets and divided them into human and environmental samples based on whether they were collected from human body parts (gut/mouth/skin/vagina) or environmental materials (soil/marine/freshwater/wastewater). The detailed information of these datasets is shown in Table 1. All shotgun sequencing data of human samples were processed with MetaPhlAn3 for quantitative species-level taxonomic profiling, and then the microbiota composition of samples from various cohorts was evaluated (Figure 1). Compositional PCA for beta diversity showed a high dispersion of the data (Figure 1A). For gut samples, PCA demonstrated the separation of samples from different cohorts (PC1: 22.03%, PC2: 4.43%). Samples of the majority of cohorts tended to cluster to the bottom left or right, indicating inter-cohort similarity in microbiota composition. An exception was PRJEB14847, samples which were far away from other cohorts and clustered into two centers, suggesting the existence of within-study clusters. The microbiota composition of mouth samples showed a high dispersion, with samples from different cohorts clustered separately into centers. The distribution of skin samples was highly concentrated and had the least variation across all habitats (PC1: 14.05%, PC2: 5.41%), whereas within-cohort vaginal samples showed a higher degree of variation, but the PCA ultimately only explains a small amount of the total variance (PC1: 16.54, PC2: 13.95%). Overall, the separation of samples across cohorts is greater than the separation within cohorts.

Abundant species are those that comprise the top 80% of the total abundance in a sample when ranked by decreasing relative abundance. The proportion of abundant species of different cohorts was consistently low, with a median of 8.15% to 21.57% (Figure 1B). This result has been previously observed in other studies, where a small number of species contributed most of the abundance in the entire microbial community [51,52]. As a function of sampling effort, the rarefaction/extrapolation (R/E) curve provides the expected number of observed/predicted abundant species, with the number of abundant species in most cohorts reaching saturation at 50 samples (Figure 1C).

### 3.2. Workflow of Constructing Reference Genomes for Specific Cohorts

The first step of the analysis workflow is the sequencing quality control to obtain high-quality reads, followed by the alignment of short reads to the genomes in the basic reference database (Figure 2). Next, reads that do not meet the post-filtering criteria are discarded: low-quality reads (MAPQ < 5), reads with non-perfect matches (insertion, deletion, skipped region, soft clipping, and hard clipping), and reads with more than three mismatches. For reads that mapped to multiple positions in the reference genome, we only kept records of the best alignment. Reference genomes were ranked based on the mapping rate (number of reads mapped to each reference genome/number of total reads), and reference genomes for each cohort were constructed depending on the number of genomes specified by users. In brief, users can easily construct cohort-specific reference genomes of any size by simply providing shotgun sequencing data.

### 3.3. Evaluation of Short-Read Sequence Aligners

The construction of cohort-specific reference genomes relies on the alignment of short sequencing reads to the basic reference database, which necessitates the inclusion of as many microbial reference sequences as feasible in the database. Large and comprehensive reference databases require more computational resources, especially memory. Therefore, we evaluated the performance of three commonly used aligners, Bowtie2 (v2.3.5.1) [53], BWA (v0.7.17) [54], and Minimap2 (v2.24) [55], during indexing and alignment. Minimap2 shows a significant speed advantage as it is about 92 times faster than Bowtie2 and 35 times faster than BWA, but at the cost of high memory usage (Table 2). BWA has the lowest memory consumption among all aligners, making it superior for indexing large reference databases.

Next, we investigated how the aligners scaled with the number of threads by running them with one, four, and eight threads as multithreading is the standard use case (Figure 3). The alignment time nearly halves for the aligners when doubling the number of threads, suggesting that the tools make efficient use of the resources. Minimap2 runs the fastest of all the aligners, but this advantage diminishes as the number of threads rises. In terms of memory usage, Bowtie2 has the lowest peak memory across all experiments. Followed by this is BWA, whose memory usage is slightly higher than Bowtie2 when using one thread but increases fast as the number of threads grows. BWA and Bowtie2 are both BWT-based aligners, whose indices may be shared by multiple tasks to reduce memory consumption. Minimap2’s memory usage is relatively high compared to memory-efficient tools Bowtie2 and BWA, which reach a peak memory usage of 15 GB when running 10 tasks simultaneously. In summary, BWA and Bowtie2 may save more memory when working with large reference genome databases, while Minimap2 is faster when using small reference databases. A large reference database could be divided into sub-databases when memory resources are restricted, and our pipeline will automatically integrate the results after mapping them individually.

### 3.4. Construction of Specific Reference Genomes for Each Dataset

Comprehensive microbial reference databases are essential for the construction of specific reference genomes; thus, we constructed two microbial reference databases, MetaPhlAn3_db_25k and Env_db_6k, applicable to human and environmental samples. MetaPhlAn3_db_25k is a microbial reference database comprising 25,435 bacterial, archaeal, and eukaryotic reference genomes. As a complement to MetaPhlAn3_db_25k, the Env_db_6k database contains 6149 reference genomes of microorganisms associated with environmental metagenomes (mainly soil and water). We designated MetaPhlAn3_db_25k as the basic reference database for human-associated samples and randomly selected 50 samples from each dataset for the construction of specific reference genomes, as 50 samples might reflect the abundance species composition of the whole cohort. To identify the optimal number of genome sequences, we investigated how the mapping rate varied with the number of genomes (Appendix A). The mapping rate of most cohorts saturated when the number of genomes reached 1000; hence, we output the top 1000 genomes as the specific reference genome for each dataset. The size of the reference genome constructed for each dataset ranged from 2.4 to 3.8 GB, which is comparable to the size of the human genome. For environment-associated samples, both the MetaPhlAn3_db_25k and Env_db_6k databases were used as the basic reference databases to construct the reference genomes of 2.8–3.9 GB in size for each dataset.

### 3.5. Performance of Reference Genome-Based Compression of Each Dataset

We evaluated the performance of reference-based compression using Genozip, a lossless compression tool designed for genomic data with both reference-based and reference-free modes [13]. Genozip was selected for its superior performance in compressing genomic sequences compared with other methods. Moreover, it supports various input and output formats, such as FASTA, FASTQ, SAM, BAM, CRAM, etc., and provides a series of extendable downstream analysis tools for high flexibility and convenience. In general, reference-based compression yielded a higher compression ratio than the general compression tool Gzip and the reference-free mode (Figure 4, Appendix A). The average compression ratio of reference-based compression for the total of 2457 human samples is 9.9, which is 2.5 times and 1.6 times better than that of Gzip and reference-free mode, respectively. When compressed using cohort-specific reference genomes, the compression ratios for all samples exceeded four, and half of the samples (1401/2457) reached eight. Over 10% (290/2457) of samples obtained a score of 15, with the best result being 29.4. The average compression ratio of each cohort ranged from 5.6 to 18.1, with an average improvement of 74–264% over Gzip (Appendix A). The low compression ratios of some datasets were potentially due to the relatively shallow sequencing depth, such as PRJEB24041 and PRJEB12449 from the human gut and PRJNA46333 from skin (Appendix A). The average compression ratio for 1693 environmental samples is 8.2, which is 1.9 times better than that of Gzip and 1.1 times higher than the reference-free method (Appendix A). Approximately 7% (120/1693) of the environmental samples were able to obtain a compression ratio of 15, with the best result being 33.8. Samples with high compression ratios were mostly from the wastewater datasets PRJNA80167 and PRJNA746354. The average compression ratios for each dataset ranged from 5.5 to 17.2, with an average improvement of 49–183% compared to Gzip.

## 4. Discussion

Novel reference-based compression tools for genomic data have been developed in recent years, but their acceptable reference genome size is usually limited, yet sensitive to the choice of reference genomes [56,57,58]. To streamline current reference databases and enhance the compression ratio of metagenomes, we introduce a pipeline for constructing a lightweight and cohort-specific reference genome using a small sample size, which can be used for compressing large cohort samples.

We collect genome sequences with high mapping rates in the reference database to create a cohort-specific reference genome, as the mapping rate has a positive correlation with the compression ratio for samples within the same cohort. Nevertheless, it should be noted that high mapping rates may not necessarily result in high compression rates when comparing across cohorts. This result is predictable as the reference genome only impacts the compression of nucleotide sequences, while the actual compression also includes header lines and quality scores. Quality scores occupy a significant chunk of the storage space and are more difficult to compress compared with other components of FASTQ files [59]. The coverage and depth of each contig can be shown by the extension tools of Genozip, which is a valuable feature that may simplify our pipeline for the construction of reference genomes. Unfortunately, we were unable to test this feature using either MetaPhlAn3_db_25k or Env_db_6k reference databases due to the limits of the free edition of Genozip regarding the size of the reference genome. In addition to the mapping rate, other factors such as the location, host information, and collection method of the samples may also affect the quality and compression efficiency of the constructed reference sequences. However, some metadata (e.g., host information and collection methods) are difficult to access due to inconsistencies in the metadata formats of different public platforms, which poses challenges for further analysis and comparison. We encourage future researchers to provide richer metadata information when uploading data to facilitate subsequent analysis and comparison.

Our findings indicate that environmental samples have lower species richness and less compression improvement than human samples when using reference genomes, as illustrated in Appendix A. This highlights the diversity and complexity of environmental microbiome communities, and the lack of adequate and comprehensive reference genomes for less-studied microbial species. To enhance the quality of reference genomes and improve habitat specificity, minimizing genomic redundancy and developing habitat-specific reference databases have proven effective. Notably, several well-curated reference genome datasets of the human microbiome have been developed in recent years, including those from low-abundance and unculturable bacteria, enriching the current reference catalog for the human microbiome [60,61,62]. Efforts have also been made to develop habitat-specific reference databases for environmental metagenomes. For example, Choi et al. developed RefSoil, a curated reference database of soil microbial genomes, and extended it to include plasmids of soil microorganisms as the RefSoil+ database [63,64]. Klemetsen et al. constructed the marine prokaryotic genome databases MarRef and MarDB using non-redundant genome and metagenome datasets obtained from ENA and NCBI [65]. These habitat-specific databases have facilitated the classification and analysis of metagenomic samples, and as these specific databases continue to be expanded and improved on, reference genomes constructed using our pipeline will become more targeted.

In summary, our work offers fresh ideas for reference-based compression and the construction of cohort-specific reference genomes. With approximately 2–4 GB of reference genomes, we achieved impressive compression ratios of 5.5–18.1 (1.5–3.6 times better than Gzip) across 29 datasets, with significant savings in storage space and cost. This is especially beneficial for large-scale genomic projects that generate and store massive amounts of data. Our method reduces the I/O time for large files and improves the efficiency and scalability of genomic data processing and analysis pipelines. Furthermore, our approach is adaptable to improvements in storage systems, which can enhance data access speed. This, in turn, facilitates rapid data retrieval and sharing among researchers and clinicians.

While our method has demonstrated impressive advantages in reference-based compression, it is important to acknowledge its limitations and consider future directions for improvement. Although the size of the microbial reference genome is already approaching that of the human genome, there is still potential for further compression by removing redundant regions based on their coverage degree. It is also important to test our method on third-generation sequencing data, which may have different characteristics and requirements for compression compared to NGS short reads. A significant challenge is to improve the availability and quality of reference genomes in the database, especially for less-studied microbial species, which may affect the compression efficiency for samples collected from environmental sources. As genome databases continue to evolve and improve, we anticipate that the quality and specificity of cohort-specific databases will enhance, further strengthening the effectiveness and applicability of our method.

## Figures and Tables

**Figure 1 microorganisms-11-02560-f001:**
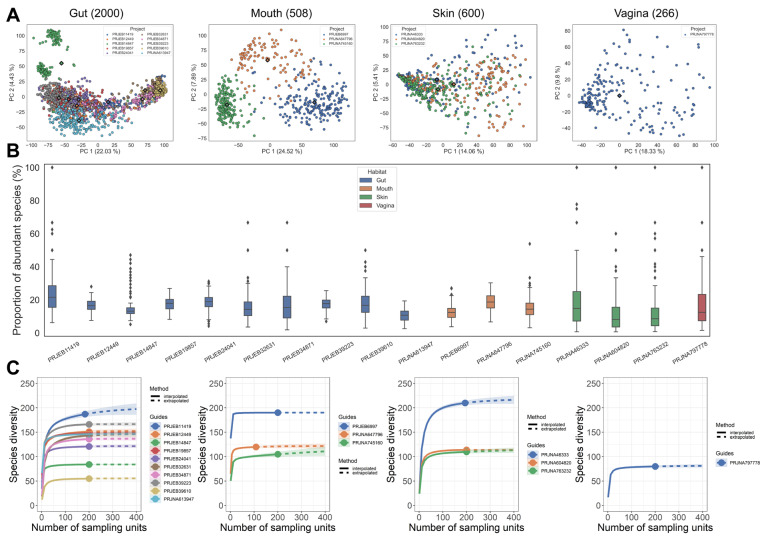
Microbiota composition analysis of human samples from various cohorts. (**A**) PCA demonstrates the clustering of samples from different cohorts based on the CLR-transformed species relative abundance data of various cohorts. The diamonds indicate cluster centroids. Number in parentheses represents the sample size. (**B**) Boxplot illustrates the proportion of abundant species in each sample. Abundance species are those that made up the top 80% of the total abundance in a sample. Each dot represents a sample from that cohort. Outliers are shown as dots beyond the whiskers of the boxplot, which reflect samples that deviate significantly from the normal range of abundant species proportions. Number in parentheses represents the sample size. (**C**) Sampling-unit-based rarefaction and extrapolation (R/E) curves of abundant species of each cohort. The dots indicate the actual number of specimen records and separate the interpolated (solid line) from extrapolated (dashed line) regions of each curve. The shaded areas represent 95% confidence intervals (based on a bootstrap method with 100 replications). Number in parentheses represents the sample size.

**Figure 2 microorganisms-11-02560-f002:**
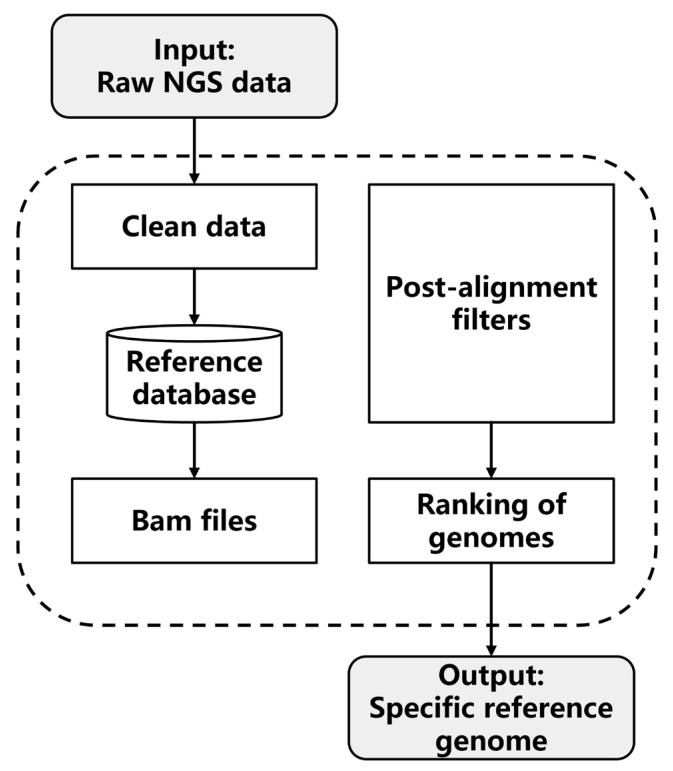
Workflow of reference genome construction. The next-generation sequencing (NGS) reads of input samples were trimmed and then mapped to the reference database. Post-alignment filters were used to remove low-quality reads, characterized by sequencing errors, small variants (indels and SNPs), and multiple mismatches. Genomes in the reference database were ranked based on the matching rate, and the final specific reference genome was constructed according to the output numbers of genomes specified by users.

**Figure 3 microorganisms-11-02560-f003:**
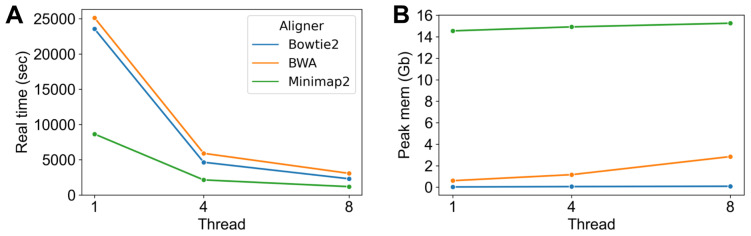
Real time and peak memory in multithread mode. (**A**) The real time (in seconds) of three aligners (Bowtie2, BWA and Minimap2) with different number of threads (1, 4 and 8). (**B**) The peak memory (in GB) of three aligners with different number of threads.

**Figure 4 microorganisms-11-02560-f004:**
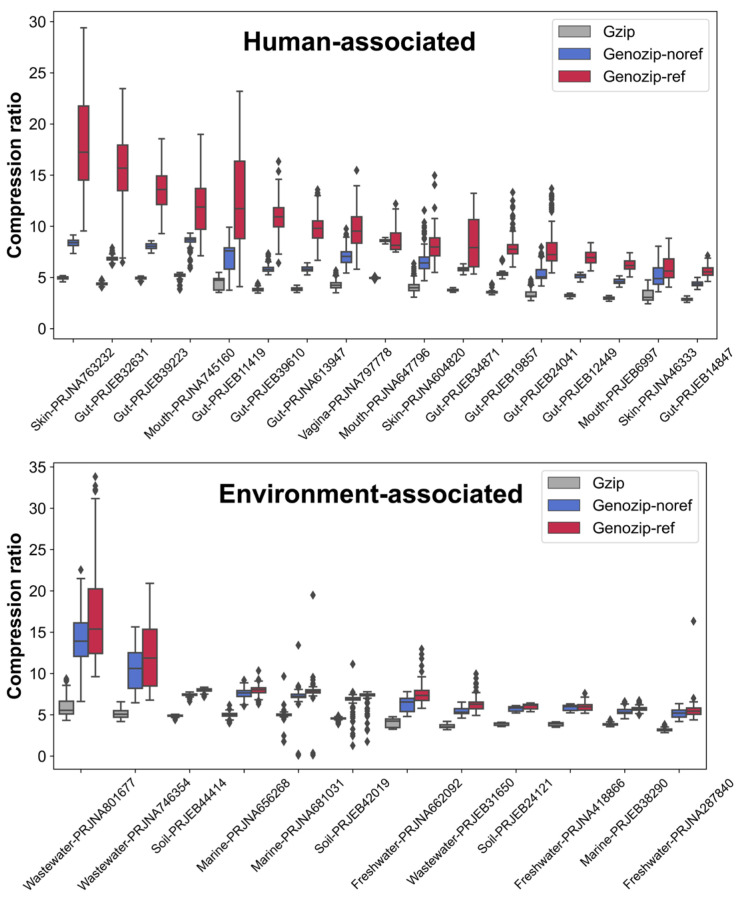
Compression ratio of samples from 29 cohorts under different compression tools and modes. The gray, blue, and red boxes represent the compression ratios of samples compressed with Gzip, Genozip without reference, and Genozip with reference, respectively. Compression ratio is defined as the ratio of the original file size to the compressed file. Outliers are shown as dots beyond the whiskers of the boxplot.

**Table 1 microorganisms-11-02560-t001:** Summary of the metagenomic datasets used in this study.

Habitats	BioProject	Sample Size	Data Size	Sample Used in This Study	References
Human gut	PRJEB11419	39,038	1003.18 Gb	200	[20]
	PRJEB12449	882	371.38 Gb	200	[21]
	PRJEB14847	372	1.18 Tb	200	[22]
	PRJEB19857	350	520.69 Gb	200	[23]
	PRJEB24041	633	480.72 Gb	200	[24]
	PRJEB32631	1679	1.34 Tb	200	[25]
	PRJEB34871	1197	1.13 Tb	200	[26]
	PRJEB39223	2196	2.97 Tb	200	[27]
	PRJEB39610	644	974.51 Gb	200	[28]
	PRJNA613947	348	871.69 Gb	200	[29]
Human mouth	PRJNA745160	888	657.43 Gb	200	[30]
	PRJNA647796	108	219.63 Gb	108	[31]
	PRJEB6997	530	2.01 Tb	200	[32]
Human skin	PRJNA46333	8774	1.87 Tb	200	[33]
	PRJNA604820	516	269.29 Gb	200	[34]
	PRJNA763232	289	210.76 Gb	200	[35]
Human vagina	PRJNA797778	542	649.33 Gb	200	[36]
Soil	PRJEB42019	7557	726.36 Gb	200	[37]
	PRJEB24121	290	180.83 Gb	200	[38]
	PRJEB44414	195	239.80 Gb	195	[39]
Marine	PRJNA656268	1942	1.16 Tb	200	[40]
	PRJNA681031	305	268.79 Gb	200	[41]
	PRJEB38290	308	1.11 Tb	200	[42]
Freshwater	PRJNA287840	729	154.65 Gb	200	[43]
	PRJNA662092	184	265.07 Gb	184	[44]
	PRJNA418866	790	335.47 Gb	116	[45]
Wastewater	PRJNA746354	3188	447.20 Gb	200	[46]
	PRJNA801677	2105	452.38 Gb	200	[47]
	PRJEB31650	567	689.20 Gb	200	[48]

**Table 2 microorganisms-11-02560-t002:** Indexing time and peak memory of indexing for aligners using one thread.

Aligner	Ref1000 (3.8 GB)	
	Time (s)	Mem (GB)
Bowtie2	15,081	10
BWA	5821	6
Minimap2	164	25

## Data Availability

Open-source software was used for analysis, including the Aspera ascp command line client (v3.9.1) for data downloading, fastp (v0.23.1) and MetaPhlAn3 for data pre-processing, Python and R package iNEXT for data analysis, BWA for data evaluation, and Genozip (v13.0.20) for data compression test. Our pipeline can be freely download at https://github.com/wanglinqi123/MetaRef (accessed on 20 March 2022). We provide the Accession number of microbial genomes for MetaPhlAn3_db_25k and Env_db_6k databases on Github, and users can build these two base databases by themselves using scripts. The download address is https://github.com/wanglinqi123/MetaRef/tree/main/BasicRefDB (accessed on 20 March 2022).

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
