# Peer review of "A Pipeline for Constructing Reference Genomes for Large Cohort-Specific Metagenome Compression"

_microorganisms, 2023, doi:10.3390/microorganisms11102560_

Round 1

Reviewer 1 Report

Wang et al. proposed a novel approach for constructing research-specific microbial reference genomes, which utilizes published human and environmental metagenomic sequencing data to compress raw sequencing data. The procedure is as follows: representative microbial genomes with high abundance are selected based on the metagenomic data from published research. Subsequently, a reference sequence set specific to the sample set is constructed through simple filtering and splicing. Published open-source software is employed for comparison and compression. Overall, this research idea is straightforward and has improved the data compression rate, thereby providing important insights for further development in this field. In principle, I support the publication of this paper in this journal. Nevertheless, I would like to highlight a few minor concerns that the author should address.

Question 1: I recommend that the author integrates a ready-to-use suite of scripts or tools, particularly those based on snakemake or other automated processes, to assist in implementing this idea.

Question 2: Although the microbial reference genome presented in this paper already approaches the size of the human genome, is there still potential for further compression? For instance, it would be useful to explore if different regions of the genome can be ranked based on coverage degree, as determined by existing data comparison results. This could involve setting retention ratios or coverage depth thresholds and providing users with the option to choose.

Question 3: Have you considered incorporating the concept of K-mer frequency into your research?

Question 4: How can the built reference sequence group be dynamically updated? It would be valuable to discuss potential approaches for achieving this.

Other: In Figure 1c, it is recommended to directly use abbreviated sample names rather than detailed dataset IDs for improved clarity.

Author Response

Dear reviewer,

Thank you for giving us an opportunity to revise our manuscript. Your valuable suggestions have helped us improve the manuscript further! We have addressed all the comments and please see the point-by-point responses below.

Question 1: I recommend that the author integrates a ready-to-use suite of scripts or tools, particularly those based on snakemake or other automated processes, to assist in implementing this idea.

Response: Thank you for your suggestion. We have wrapped the whole pipeline into a Snakemake-based workflow that automates all the steps. Users only need to prepare a sample list and run the work.sh script in the working directory to construct the specific reference genome. We have updated the quick start user manual on our GitHub page, please see https://github.com/wanglinqi123/MetaRef.

Question 2: Although the microbial reference genome presented in this paper already approaches the size of the human genome, is there still potential for further compression? For instance, it would be useful to explore if different regions of the genome can be ranked based on coverage degree, as determined by existing data comparison results. This could involve setting retention ratios or coverage depth thresholds and providing users with the option to choose.

Response: It is really true as the reviewer suggested that coverage-based approaches can take advantage of the high degree of similarity between multiple genomes and reduce redundant information in the reference genome. We believe this is a promising approach to help further reduce the size of microbial reference genomes, making them more suitable for large-scale cohort compression. We also agree that setting a retention ratio or coverage depth threshold is a good idea to provide users with more flexibility in choosing the right reference genome quality and size for their needs. In the discussion section, we have discussed potential directions for optimizing our method (lines 330-332). Due to the limited time frame, we plan to implement this function in a coming release.

Question 3: Have you considered incorporating the concept of K-mer frequency into your research?

Response: Thank you for your valuable suggestion. K-mer frequency has its advantages in terms of processing speed. We plan to perform K-mer analysis on the sequencing reads and reference genomes to find the most suitable reference genome combinations for the input samples. Determining the length of K-mer and selecting the best genome combination from the reference database could be challenging. We will explore this direction and update it in future releases.

Question 4: How can the built reference sequence group be dynamically updated? It would be valuable to discuss potential approaches for achieving this.

Response: Thanks for the valuable comments. We constructed the specific reference sequences based on two basic databases, MetaPhlAn3_db_25k and Env_db_6k, which contain microbial genome sequences isolated from various environments. We will update these two databases regularly to reflect the newly discovered microbial species and variations. We will also publish the version numbers on our GitHub page and change logs of the databases after each update to make it easy for users to trace and compare the results of different versions.

Other: In Figure 1c, it is recommended to directly use abbreviated sample names rather than detailed dataset IDs for improved clarity.

Response: Thank you for the suggestion. The purpose of Figure 1C is to show the rarefaction curve of abundant species in each dataset as the sample size increases, and the different color lines represent different datasets rather than samples. Therefore, we have indicated dataset IDs in the legend to facilitate the reader's understanding.

Once again, thank you for your contribution to our study. We hope the revised manuscript is now suitable for publication in Microorganisms.

Kind regards,

Yan Zhou, PhD

State Key Laboratory of Genetic Engineering

School of Life Sciences, Fudan University

Shanghai, China

E-mail: zhouy@fudan.edu.cn

Reviewer 2 Report

I checked your manuscript and described comments below..

In genome analysis, it is important to combine several programs into a pipeline for efficient analysis.

In this paper, we propose a very method by combining programs.

I think you should consider the following points.

1.       It would be better to include a flow chart in the Materials and Methods section.

2.       References 23, 31, 35, 37, 41, 42, 46, 47 are URLs/links, not references. I think it would be better to put it in the text.

I don't think this paper has any major mistakes or grammatical problems.

Author Response

Dear reviewer,

Thank you for giving us an opportunity to revise our manuscript. Your valuable suggestions have helped us improve the manuscript further! We have addressed all the comments and please see the point-by-point responses below.

Question 1: It would be better to include a flow chart in the Materials and Methods section.

Response: Thank you for the suggestion. We described the workflow in detail in the Results section (lines 176-194 and Figure 2) to provide a visual representation and enhance clarity of the research findings for readers. We appreciate the reviewer’s concern regarding the inclusion of a flow chart in the Materials and Methods section for readers interested in technical details. In response to this feedback, we have carefully considered the suggestion and have addressed it by adding a detailed description of the process for constructing reference genomes in the Materials and Methods section (lines 126-128). Thank you for bringing this to our attention, and we value your input in improving the clarity of our manuscript.

Question 2: References 23, 31, 35, 37, 41, 42, 46, 47 are URLs/links, not references. I think it would be better to put it in the text.

Response: Thank you for pointing out this issue. The URLs/Links we cite are online datasets that contain project data relevant to this study. These data sets do not have corresponding publications but are directly uploaded to public databases. Therefore, we follow the citation specification for online data in Microorganisms to make it easier for readers to access and review these data. We appreciate your feedback and will ensure that future citations are appropriately formatted.

Once again, thank you for your contribution to our study. We hope the revised manuscript is now suitable for publication in Microorganisms.

Kind regards,

Yan Zhou, PhD

State Key Laboratory of Genetic Engineering

School of Life Sciences, Fudan University

Shanghai, China

E-mail: zhouy@fudan.edu.cn

Reviewer 3 Report

Manuscript 2570713 discusses such an important element of modern biology as Metagenomic data compression, which generally corresponds to the profile of Microorganisms.

There are only minor comments that do not affect the overall positive impression of the scientific work.

- Authors should work on keywords. For example, "big data acquisition" is not used in other sections of the manuscript.

- Is it necessary to provide more specific information about the samples in the article itself, what is human and environmental? Does the main result considered by the authors affect the location of the sample, gender, perhaps the method of selection, storage, DNA extraction, etc.?

-The sentence on line 262 does not start with a capital letter, it needs to be corrected.

Author Response

Dear reviewer,

Thank you for giving us an opportunity to revise our manuscript. Your valuable suggestions have helped us improve the manuscript further! We have addressed all the comments and please see the point-by-point responses below.

Question 1: Authors should work on keywords. For example, "big data acquisition" is not used in other sections of the manuscript.

Response: Thank you so much for your careful check. We have revised the keywords to make them more consistent with our research content and objectives.

Question 2: Is it necessary to provide more specific information about the samples in the article itself, what is human and environmental? Does the main result considered by the authors affect the location of the sample, gender, perhaps the method of selection, storage, DNA extraction, etc.?

Response: Human samples refer to samples collected from human body parts (Gut/Mouth/Skin/Vagina), and environmental samples refer to samples collected from environmental materials (Soil/Marine/Freshwater/Wastewater). We are very sorry for the confusion and have clarified the definitions of human and environmental samples in lines 137-141 to provide a better understanding for readers. The quality of the constructed reference genomes, as well as the compression ratio, can be influenced by various factors such as location, host information, and collection method, which may vary among different datasets. We have provided detailed information of the samples in Supplementary Table S1, including the source, dataset ID, and sequencing platform of the samples. However, since these data were downloaded from public databases, it is difficult to obtain complete metadata such as host information and collection methods. We address this issue in the Discussion section and recommend that future researchers provide more rich metadata information when uploading data to facilitate subsequent analysis and comparison. Please see lines 293-300.

Question 3: The sentence on line 262 does not start with a capital letter, it needs to be corrected.

Response: Thanks. We have made correction according to the reviewer’s comments.

Once again, thank you for your contribution to our study. We hope the revised manuscript is now suitable for publication in Microorganisms.

Kind regards,

Yan Zhou, PhD

State Key Laboratory of Genetic Engineering

School of Life Sciences, Fudan University

Shanghai, China

E-mail: zhouy@fudan.edu.cn

Reviewer 4 Report

Summary: The manuscript presents a novel approach for constructing lightweight, cohort-specific reference genomes aimed at improving the compression ratio of metagenomic data. The methodology is robust, and the results indicate significant potential for reducing storage costs in large-scale metagenomic studies. Minor Comments: Methods (Line 97-103): A more detailed explanation about how the database was constructed and why these genomes were selected would strengthen the manuscript. Also, the introduction of MetaPhlAn3 is abrupt and lacks explanatory context. A brief description of this tool and its relevance to the study would be helpful for the reader. Software Selection: The manuscript should clarify why Genozip was chosen for this study, I assume it is because of its higher performance in comparison to other methods as per the Genozip paper. Discussion: The discussion section does well in summarizing the work but could delve deeper into certain aspects. For example: * Discussing potential limitations of your method, especially when applied to less-studied microbial communities or different types of sequencing data. * Exploring the implications of your findings for real-world applications. How do these cohort-specific reference genomes contribute to ongoing research or industrial applications?

Author Response

Dear reviewer,

Thank you for giving us an opportunity to revise our manuscript. Your valuable suggestions have helped us improve the manuscript further! We have addressed all the comments and please see the point-by-point responses below.

Question 1: Methods (Line 97-103): A more detailed explanation about how the database was constructed and why these genomes were selected would strengthen the manuscript. Also, the introduction of MetaPhlAn3 is abrupt and lacks explanatory context. A brief description of this tool and its relevance to the study would be helpful for the reader.

Response: Thank you for pointing out this problem in manuscript. We have re-written this part according to the reviewer’s suggestion. Please see lines 96-116.

Question 2: Software Selection: The manuscript should clarify why Genozip was chosen for this study, I assume it is because of its higher performance in comparison to other methods as per the Genozip paper.

Response: We gratefully appreciate for your valuable suggestion. We chose Genozip for its superior performance in compressing genomic sequences compared with other methods. This tool also supports both reference-based and reference-free compression modes, as well as various input and output formats. We have provided a detailed explanation of our choice of Genozip in the revised version of the manuscript, please see lines 245-250.

Question 3: Discussion: The discussion section does well in summarizing the work but could delve deeper into certain aspects. For example: * Discussing potential limitations of your method, especially when applied to less-studied microbial communities or different types of sequencing data. * Exploring the implications of your findings for real-world applications. How do these cohort-specific reference genomes contribute to ongoing research or industrial applications?

Response: Thank you for the above suggestions. We have added the potential limitations and applications of our method in the discussion section, please see lines 319-339. We hope that these additions will make our discussion section more comprehensive and informative for the readers. Thanks again for the constructive suggestions and your efforts in improving the quality of our manuscript.

Once again, thank you for your contribution to our study. We hope the revised manuscript is now suitable for publication in Microorganisms.

Kind regards,

Yan Zhou, PhD

State Key Laboratory of Genetic Engineering

School of Life Sciences, Fudan University

Shanghai, China

E-mail: zhouy@fudan.edu.cn